# Impact of Aβ40 and Aβ42 Fibrils on the Transcriptome of Primary Astrocytes and Microglia

**DOI:** 10.3390/biomedicines10112982

**Published:** 2022-11-19

**Authors:** Xiaoyue Zhu, Joseph M. Schrader, Brandon A. Irizarry, Steven O. Smith, William E. Van Nostrand

**Affiliations:** 1George and Anne Ryan Institute for Neuroscience, University of Rhode Island, Kingston, RI 02881, USA; 2Department of Biomedical and Pharmaceutical Sciences, University of Rhode Island, Kingston, RI 02881, USA; 3Center for Structural Biology, Department of Biochemistry and Cell Biology, Stony Brook University, Stony Brook, NY 11794, USA

**Keywords:** amyloid β-protein, fibrils, astrocytes, microglia, RNA sequencing

## Abstract

Fibrillar amyloid β-protein (Aβ) deposits in the brain, which are primarily composed of Aβ40 or Aβ42 peptides, are key pathological features of Alzheimer’s disease (AD) and related disorders. Although the underlying mechanisms are still not clear, the Aβ fibrils can trigger a number of cellular responses, including activation of astrocytes and microglia. In addition, fibril structures of the Aβ40 and Aβ42 peptides are known to be polymorphic, which poses a challenge for attributing the contribution of different Aβ sequences and structures to brain pathology. Here, we systematically treated primary astrocytes and microglia with single, well-characterized polymorphs of Aβ40 or Aβ42 fibrils, and performed bulk RNA sequencing to assess cell-specific changes in gene expression. A greater number of genes were up-regulated by Aβ42 fibril-treated glial cells (251 and 2133 genes in astrocyte and microglia, respectively) compared with the Aβ40 fibril-treated glial cells (191 and 251 genes in astrocytes and microglia, respectively). Immunolabeling studies in an AD rat model with parenchymal fibrillar Aβ42 plaques confirmed the expression of *PAI-1*, *MMP9*, *MMP12*, *CCL2*, and *C1r* in plaque-associated microglia, and *iNOS*, *GBP2*, and *C3D* in plaque-associated astrocytes, validating markers from the RNA sequence data. In order to better understand these Aβ fibril-induced gene changes, we analyzed gene expression patterns using the Ingenuity pathway analysis program. These analyses further highlighted that Aβ42 fibril treatment up-regulated cellular activation pathways and immune response pathways in glial cells, including *IL1β* and *TNFα* in astrocytes, and microglial activation and *TGFβ1* in microglia. Further analysis revealed that a number of disease-associated microglial (DAM) genes were surprisingly suppressed in Aβ40 fibril treated microglia. Together, the present findings indicate that Aβ42 fibrils generally show similar, but stronger, stimulating activity of glial cells compared with Aβ40 fibril treatment.

## 1. Introduction

Cerebral aggregation of fibrillar amyloid-β (Aβ) peptides is a hallmark of Alzheimer’s disease and related disorders (ADRD). Proteolytic cleavage within the transmembrane region of the amyloid precursor protein generates Aβ peptides of various lengths, of which ~90% are comprised of Aβ40 (40 amino acid residues in length), and 5-10% of Aβ42 [1]. The characteristic cored senile Aβ plaques in AD are composed primarily of Aβ42, and increases in the Aβ42/40 ratio are associated with early onset AD [1,2,3]. On the other hand, cerebral amyloid angiopathy (CAA) is a cerebral small vessel disease arising from the deposition of Aβ fibrils in the cortical and leptomeningeal blood vessels rather than in parenchymal plaques [4,5,6]. Though it occurs in >80% of AD cases as the most common comorbidity, CAA also exists in the absence of AD, occurring sporadically in >60% of people over the age of 80 years [4,5,7,8,9]. While both Aβ40 and Aβ42 are present in CAA vascular amyloid deposits, CAA is primarily associated with Aβ40, in contrast to senile plaques in AD that are predominantly composed of Aβ42 [4,10,11,12]. Despite the significant clinical burden of both AD and CAA, underlying mechanisms of disease pathogenesis are still poorly understood, and effective therapeutic options are lacking.

In the progressive pathogenesis of AD, extracellular Aβ accumulation increases in the brain, resulting in neuroinflammation indicated by astrogliosis, microglial activation, and dystrophic neurites [13,14]. The innate immune cells involved in these processes are primarily microglia and astrocytes, and neuroinflammation driven by microglia and astrocytes activated by Aβ42 senile plaques is now considered a critical mechanism contributing to AD pathology [15,16,17,18,19]. On the other hand, CAA is marked clinically by intracerebral hemorrhages, white matter hyperintensities, cerebral infarction, and microbleeds, as primarily fibrillar Aβ40 accumulation in the walls of cerebral vessels drives a loss of endothelial/vascular integrity [4,5,8,20,21,22,23]. Additionally, dyshoric fibrillar Aβ projections into the brain parenchyma leads to significant perivascular neuroinflammation [24]. As in AD, CAA-related inflammation is associated with activation of both astrocytes and microglia [25,26,27]. Thus, neuroinflammation mediated by astrocytes and microglia appears to be a critical component of the pathogenesis of ADRDs, and can be driven by both Aβ40 and Aβ42 peptides. 

Astrocytes are involved in regulation of cerebral blood flow, maintenance of synaptic homeostasis, and neurotrophic support for synapses [28,29,30,31,32,33,34,35]. Additionally, astrocytes can form unique perivascular channels in the CNS, which eliminate neurotoxic waste products, including Aβ fibrils and tau species [36,37,38]. On the other hand, microglia survey the CNS microenvironment, sensing damage signals and supporting neuronal network structures [14,39]. Through immune surveillance, microglia recognize and remove pathogens, cell debris, or abnormal proteins (including Aβ peptides) via pinocytosis, phagocytosis, or receptor-mediated endocytosis [40]. Both astrocytes and microglia are recognized to be activated to contribute to neuroinflammatory processes [41,42]. These activated glia cells lose their homeostatic functions, reducing their secretion of neurotrophic factors and producing increased amounts of proinflammatory cytokines and chemokines, which could aid in pathogen or toxin clearance, but also induce neuronal dysfunction and damage [41,42]. 

However, how glial cells are activated by and respond to specific forms of Aβ fibrils remains unclear. While both Aβ40 and Aβ42 peptides are present in both parenchymal plaque and vascular lesions, it is currently unknown whether Aβ40 and Aβ42 fibrils exhibit distinct contributions to pathology in ADRD progression. Aβ40 and Aβ42 fibrils share a characteristic cross-β sheet structure with other amyloid fibrils [43,44], and multiple studies have reported that, structurally, Aβ fibrils are highly polymorphic [45,46,47,48,49]. They have been described as protofilaments intertwined in a helical geometry, with varying width and helical pitch, different cross-section profiles, and different interactions between the protofilaments. The arrangement of Aβ peptides within the fibril can vary drastically between different polymorphs, with potential implications for biological activity. Thus, elucidating the structural details of specific Aβ fibrils is an important first step toward understanding their pathological activity, which can further guide the design of therapeutic strategies to diagnose, treat, or even prevent ADRD. 

Previous investigations of Aβ treatment impacts on gene expression in glial cells have largely focused on a single cell type, a single Aβ fibril species, or a mixture of Aβ fibrils of varying lengths [18,19,36,37,40,50,51,52]. In contrast with these previous studies, herein, we conducted the first comparative study of the biological responses of astrocytes and microglia to homogeneous preparations of Aβ40 and Aβ2 fibrils, the predominant fibrillar species in the brains of patients with ADRD. Templated-growth was used to isolate homogenous populations of Aβ40 and Aβ42 fibrils, eliminating non-fibrillar Aβ, and a combination of biophysical techniques was used to characterize their structures. Primary cultures of rat astrocytes and rat microglia were exposed to each type of Aβ fibril, and the resulting transcriptomic changes and gene expression profiles were characterized using RNA sequencing. Subsequent pathway analysis of transcriptomic results was performed with the Ingenuity pathway analysis (IPA) software program (Version: 81348237, Qiagen Inc., Redwood City, CA, USA) to provide mechanistic context. While Aβ42 produced stronger responses in general in each cell type, both Aβ fibrils produced common and distinctive changes impacting the activation states and functional responses in the treated astrocytes and microglia. Thus, we report new information on the specific activation of astrocytes and microglia by defined populations of Aβ40 and Aβ42 fibrils, harboring important insight into the contributions of these fibrillar species to neuroinflammation in ADRD. 

## 2. Materials and Methods

### 2.1. Fibril Preparation

Unlabeled and ^13^C-labeled Aβ40 and Aβ42 were synthesized using solid-phase chemistry (ERI-Amyloid; Waterbury, CT, USA) and purified by high-pressure liquid chromatography. Matrix-assisted laser desorption or electrospray ionization mass spectrometry were used to determine the peptide molecular weights, which were found to be consistent with the calculated mass for each peptide. Analytical reverse phase high-pressure liquid chromatography and mass spectrometry indicated that the purity of the peptides was 95–99%.

The initial fibrils were prepared by dissolving 0.5 mg of Aβ40 or Aβ42 in 50 µL of 50 mM NaOH for 1 h at room temperature. The samples were resuspended in 1 mL of 10 mM sodium phosphate buffer for a final concentration of 100 µM Aβ at 4 °C, and then filtered with 0.2 µm filters to yield monomeric Aβ. For the purpose of generating the initial Aβ40 fibril seeds, the Aβ monomers were sonicated using a 5/64″ diameter probe tip (Model Q125A, Qsonica, Newtown, CT, USA) at 4 °C 3× at 10% power for 30 s, with 1 m intervals. Following sonication, the sample was warmed to room temperature for 4 h, and then sonicated again 3× at 30% power for 30 s in 1 m intervals. Following sonication, the samples were incubated overnight under quiescent conditions. These fibrils were designated as generation 1. Subsequent generations were prepared by sonicating the parental fibrils, then introducing freshly prepared Aβ40 monomer at 15% (*w*/*w*) fibril seeds to monomer, and allowing them to incubate at room temperature under quiescent conditions. For both Aβ40 and Aβ42, the relatively gentle conditions for generating fibril seeds were selected in order to produce isolated fibrils with a twisted ribbon morphology, rather than the striated ribbons, composed of laterally associated fibrils, that are typically associated with stronger sonication or shaking conditions [53], and which we find to be more heterogenous. The quiescent templated-growth conditions used for propagating the fibrils retain the twisted ribbon fibril morphology. The sonication procedure fragments the fibrils, producing fibril seeds that rapidly deplete monomeric and oligomeric Aβ species. In addition, the fibrils were pelleted prior to structural measurements in order to remove soluble Aβ oligomers.

### 2.2. Transmission Electron Microscopy 

The morphology of Aβ40 and Aβ42 fibrils was assessed with negative stain transmission electron microscopy. Solutions of the fibrils were diluted and deposited onto carbon-coated copper mesh grids, then negatively stained with 2% (*w*/*v*) uranyl formate. After removing excess stain, the sample grids were allowed to air dry. The TEM micrographs were obtained with a FEI Tecnai 12 BioTwin 80 kV transmission electron microscope and an Advanced Microscopy Techniques camera (Hillsboro, OR, USA).

### 2.3. FTIR Spectroscopy

Templated growth of the Aβ40 and Aβ42 fibrils used in these studies was monitored by Fourier transform infrared spectroscopy (FTIR), using a Bruker Vertex 70v spectrometer with an attenuated total reflectance (ATR) accessory. Solutions of Aβ monomers with and without fibrils seeds were layered on a 4 mm germanium ATR plate (Pike Technologies, Fitchburg, WI, USA) using 50 µL of the peptide solution (~25 µg), and then dried with a stream of compressed air. The spectral resolution was 4 cm^−1^, and 500 scans were averaged per spectrum. The spectra were normalized to the intensity of the amide II absorbance band (~1530–1550 cm^−1^).

### 2.4. NMR Spectroscopy

Room temperature solid-state MAS NMR experiments were performed at a ^13^C frequency of 125 MHz on a Bruker AVANCE spectrometer, using 4 mm MAS probes. The MAS spinning rate was set to 10 KHz (±5 Hz). Ramped amplitude cross polarization was used with a contact time of 2 ms. The ^13^C field strength was 54.4 kHz, and the ramped ^1^H field was centered at approximately 50 kHz. Two-pulse phase-modulated decoupling was used during the evolution and acquisition periods, with a radiofrequency field strength of 82.7 kHz. Internuclear ^13^C–^13^C distance constraints were obtained from 2D dipolar assisted rotational resonance (DARR) NMR experiments [54], with a mixing time of 600 ms. Each data set contained 64 t_1_ increments and 1024 complex t_2_ points with spectral widths of 27.7 kHz in both dimensions. A total of 512 scans were averaged per t_1_ increment.

All ^13^C solid-state MAS NMR spectra were externally referenced to the ^13^C resonance of neat TMS at 0 ppm at room temperature. Using TMS as the external reference, we calibrated the carbonyl resonance of solid glycine at 176.46 ppm. The chemical shift difference between ^13^C of DSS in D_2_O relative to neat TMS is 2.01 ppm.

### 2.5. Animals

All work with animals was approved by the University of Rhode Island Institutional Animal Care and Use Committee, and in accordance with the United States Public Health Service’s Policy on Humane Care and Use of Laboratory Animals. The study was also in compliance with the ARRIVE guidelines. Timed pregnant Sprague–Dawley (SD) rats were obtained from Taconic Laboratories for glial cell culture preparations. TgF344-AD rats express mutant human amyloid precursor protein (APP_sw_) and presenilin 1 (PS1ΔE9) genes, and develop age-dependent parenchymal amyloid plaques in the cortex and hippocampus [55]. All rats were housed in a controlled room (22 ± 2 °C and 40–60% humidity) on a standard 12 h light cycle. Rat chow and water were available ad libitum.

### 2.6. Primary Rat Microglia and Astrocyte Isolation and Culture

P2 neonatal SD rat pups were used for glial cells isolation according to published protocols [56], with minor modifications. Briefly, P2 rat pups were first rinsed with 70% ethanol, then quickly decapitated with sterile sharp scissors. The heads were immediately placed into 70% ethanol. The entire brain was removed from the head and placed into a sterile dish with L15 solution (Leibowitz L15 + 0.1% BSA + 1% Pen/Strep) on ice. The cortical tissues were dissected under a stereomicroscope, and meninges were peeled off gently. Isolated cortical hemispheres were cut into small pieces and transferred to Hank’s Balanced Salt Solution (HBSS) with 1% trypsin. The tissues solution was incubated at 37 °C for 30 min, followed by the centrifugation at 50 rcf for 5 min to pellet cortical tissues. The supernatant was removed by aspiration, and the cell pellet was pipetted up and down 10× with glia cell culturing medium (Dulbecco’s Modified Eagle Medium (DMEM) + 10% fetal bovine serum (FBS) + 1% Pen/Strep). Cell suspensions were then flushed through a cell strainer (100 µm pores) and centrifuged at 1600 rcf for 5 min to collect the cell pellet. The final cell pellet contained mixed glial cells. Cells were seeded at a density of 10^7^ cells per T75 flask and incubated in a 5% CO_2_ incubator at 37 °C, with medium changed every three days. After the glial cultures were completely confluent, the flasks were placed on a shaker (Excella E24 Incubator Shaker Series) at 100 rpm at 37 °C for 1 h. The medium was carefully collected without disturbing the astrocyte layer on the flask surface, and then centrifuged at 200 rcf for 10 min. The resulting cell pellet, containing high purity microglia, was resuspended in microglia plating media (DMEM + 10% FBS + 1% Pen/Strep) and seeded in 12-well plates (#354470; Corning, Glendale, AZ, USA) at a density of 0.5 × 10^6^ cells per well. The T75 flasks continued to be shaken at 240 rpm for 6 h to remove any oligodendrocyte precursor cells. The supernatant was discarded, and the astrocyte layer was rinsed twice with PBS and incubated with trypsin-EDTA buffer. The detached astrocytes were centrifuged and re-suspended in culture medium, then seeded in 12-well plates at the density of 0.4 × 10^6^ cells per well. Isolated primary microglia and astrocytes were cultured to attach overnight. The culture media were replaced with serum-free medium (DMEM + 0.1% BSA + 1% Pen/Strep) for 3 h, then treated with or without 25 µM A 40 or Aβ42 fibrils at 37 °C for 24 h.

### 2.7. RNA Library Preparation and Sequencing 

RNA isolation, preparation, and sequencing were carried out by Omega Bioservice Company (Norcross, GA, USA) according to the TruSeq Stranded mRNA Library Prep Kit (Illumina, Inc., San Diego, CA, USA) protocol (TruSeq Stranded mRNA. Available online: https://support.illumina.com/downloads/truseq-stranded-mrna-reference-guide-1000000040498.html) (accessed on 16 November 2022). High quality poly mRNA (RINe > 8) was used for the subsequent sequencing analysis. The libraries were quantified, normalized, pooled, and subjected to cluster, and pair read sequencing was performed for 150 read cycles on a HiSeqX10 instrument (Illumina, Inc. San Diego, CA, USA), according to the manufacturer’s instructions. 

### 2.8. RNA Sequence Analysis

Data were analyzed with Illumina Basespace app suite (Available online: https://www.illumina.com/products/by-type/informatics-products/basespace-sequence-hub/apps/rna-seq-alignment.html) (accessed on 16 November 2022). The sequencing reads were aligned to the Rattus norvegicus UCSC rn5 using the STAR aligner. Salmon was used for quantification of reference genes and transcripts. The Strelka Variant caller identifies single nucleotide variants (SNVs) and small indels, and generates output in a VCF file. Manta was used to detect gene fusions. GC and mean coverage information for every target was computed using Picard. The gene counts and differential expression results were produced by DESeq2. 

### 2.9. Immunolabeling of Rat Brain Tissues

Fifteen-month-old rTgF344-AD rats were perfused with PBS, followed by 4% paraformaldehyde (PFA). Dissected brains were fixed with 4% PFA overnight, then transferred to 30% sucrose at 4 °C for three days. Tissues were embedded at the optimal cutting temperature (OCT 4585; Fisher Healthcare, Houston, TX, USA), and horizontally sectioned at 15 µm. The rat brain sections were blocked in Superblock blocking buffer (#37518; Thermo Fisher. Rockford, IL, USA) containing 0.3% Triton X-100 at room temperature for 30 min. Primary antibodies were applied overnight in a humid chamber at room temperature. Alexa Fluorescent 594-, 488- or 350-conjugated secondary antibodies were applied for 2 h at room temperature, 1:1000. 

In order to detect astrocyte markers, transgenic AD rat brain sections were incubated with rabbit antibody to C3 (1:200, AF2655, Novus, Centennial, CO, USA), and rabbit antibody to GBP2 (1:200, ab203238, Abcam, Boston, MA, USA). In order to detect microglia markers, sections were incubated with rabbit antibody to MMP9 (1:200, A0289, Abclonal, Woburn, MA, USA), rabbit antibody to MMP12 (1:100, NBP2-67344, Novus, Centennial, CO, USA), rabbit antibody to C1R (1:100, A6360, Abclonal, Woburn, MA, USA), rabbit antibody to PAI-1(1:100, NBP1-19773, Novus, Centennial, CO, USA), or rabbit antibody to CCL2 (1:200, NBP1-07035, Novus, Centennial, CO, USA). Other antibodies included mouse monoclonal antibody to human Aβ 66.1, goat polyclonal antibody to astrocyte cell marker GFAP (1:200, NB100-53809, Novus, Centennial, CO, USA), and goat polyclonal antibody to microglia cell marker Iba1 (1:200, NBP1-06014, Novus, Centennial, CO, USA). 

## 3. Results

### 3.1. Preparation and Characterization of Aβ40 and Aβ42 Fibrils

Templated growth of Aβ fibrils from peptide monomers has the advantage of generating homogeneous fibrils and eliminating non-fibrillar Aβ species [53,57]. The Aβ40 or Aβ42 fibrils used for our studies were prepared by forming fibril seeds and then using these seeds to propagate homogeneous fibril populations via several rounds of templated growth with the addition of monomeric Aβ40 or Aβ42 peptides. The seeding process was monitored by FTIR spectroscopy and negative stain TEM. The amide I and amide II vibrations in FTIR spectra serve as markers for the transition to mature fibrils (Figure 1A,B). The appearance of a sharp amide I vibration at ~1627 cm^−1^ and a shift of the amide II vibration to ~1545–1550 cm^−1^ are diagnostic of the conversion of non-fibrillar Aβ to fibrils. The lack of these features in monomeric Aβ samples incubated in the absence of seeds indicates that the Aβ monomer did not form fibrils via self-nucleation. Negative stain TEM micrographs reveal homogeneous populations of Aβ40 and Aβ42 fibrils generated by templated growth (Figure 1C,D). In both cases, the quiescent templated-growth conditions generate fibrils comprised of twisted ribbons with uniform widths of ~10 nm and cross-over distances of ~120–150 nm.

Selective ^13^C solid-state NMR measurements were conducted on the mature Aβ40 and Aβ42 fibrils in order to assess their global fold (Appendix A). NMR structural studies on mature Aβ40 fibrils formed in solution often revealed a two-stranded structure, in which Phe19 on the first strand packs against Leu34 on the second strand [55]. Solid-state NMR measurements of Aβ40 containing ring^13^CPhe19 and U-^13^C-Leu34 exhibited cross peaks between the terminal leucine methyl group and aromatic phenylalanine ring carbons, consistent with the two-stranded structure (Appendix A). In contrast, NMR [58,59] and cryo-EM [43] structural studies on Aβ42 fibrils typically showed a three-stranded structure. In order to probe for this structure, ring-^13^C-Phe20 and 3-^13^C-Ala30, as well as 2-^13^C-Gly29 and U-^13^C-Ile41, were incorporated in the Aβ42 sequence. These pairs of residues form points of contact between the first and second β-strands, as well as the second and third β-strands, respectively. The strong cross peaks between these ^13^C labels in the 2D NMR spectrum of Aβ42 are diagnostic of the three-stranded structure, which is observed in both solution [58,60] and brain-derived fibrils [61] (Appendix A). 

### 3.2. Glia Cell Purification and RNA Sequencing Transcriptional Profiling

In order to investigate how these well-characterized Aβ fibrils stimulate glial cells, we performed RNA sequencing on primary astrocytes and microglia treated with Aβ40 and Aβ42 fibrils. Primary cultured astrocytes and microglia from postnatal day 2 (P2) rat brains were immunolabeled with the astrocyte marker GFAP or microglial marker Iba-1, respectively, to demonstrate the high purity of each cell type (Appendix A). Primary glial cells were then treated with the homogeneous Aβ40 or Aβ42 fibrils at a concentration of 25 µM for 24 h, and the glial cells were then collected and processed for RNA sequencing. By examining the expression of well-described markers of major CNS cell types, astrocytes, microglia, endothelial cells, neurons, and RNA sequencing profiles showed high enrichment in astrocyte and microglial-specific genes, as well as an absence of other cell-type-specific genes (Figure 2).

### 3.3. Comparison of Glial Cell Transcripts with Aβ Fibril Treatments 

The influence of Aβ42 and Aβ40 fibrils on expression profiles in glial cells was compared to the expression profiles in the control group. The results showed that Aβ42 and Aβ40 fibrils promoted similar effects on astrocytes and microglia compared with the untreated control cultures. As shown in a Pearson correlation analysis (Figure 3, Appendix A), the r values between Aβ42 and Aβ40 fibril treatments were all above 0.9 in both astrocytes and microglia, while the r values between Aβ fibrils and control cultures in both glial cell types were around 0.6, suggesting that astrocytes and microglia undergo a great number of Aβ fibril-induced changes in gene expression.

In order to elucidate gene changes induced by specific Aβ fibril treatments, we conducted comparative differential gene expression analysis of Aβ42 and Aβ40 fibril-treated glial cells. Significantly up-regulated genes were defined as those with ≥5-fold expression compared to untreated control cell expression, with a statistical significance cutoff set at *p* < 0.05, calculated by a Student’s *t*-test. Aβ42 fibril treatment of astrocytes resulted in a greater number of up-regulated genes than Aβ40, with 251 and 191 up-regulated genes induced by Aβ42 and Aβ40 fibril treatments, respectively (Figure 4A). Among the up-regulated genes in the astrocytes, 79 were common to both Aβ40 and Aβ42 fibril treatments (Figure 4A). In order to highlight some of the important specific and common expression changes induced by the fibril treatments, we ranked the altered genes according to average abundance among all treatment groups. We also compared the relative expression of the 20 most abundant genes specifically elevated (≥5-fold control expression) in each Aβ fibril treatment, as well as those common to both. These are depicted in the heat maps in Figure 4B, along with the corresponding expression in the other treatment group. Notable Aβ42 specific up-regulated genes include *Cxcl2* and *Gbp2*. The *Cxcl2* gene codes for chemokine (C-X-C motif) ligand 2, also known as macrophage inflammatory protein two, a proinflammatory chemokine produced and released by astrocytes mediating neuroinflammatory responses [62,63]. Gbp2 is a known marker for A1 neuroinflammatory/neurotoxic astrocytes [64,65]. Meanwhile, *C3*, *Cfb*, and *Serping1*, all markers of A1 reactive astrocytes [66,67], were all commonly up-regulated in both Aβ42 and Aβ40 fibril-treated astrocytes. Additionally, the inducible neurotoxic factor *Lcn2*, which is produced and secreted in reactive astrocytes [68], was commonly up-regulated in both fibril treatment groups. Interestingly, *SphK1*, which codes for sphingosine kinase 1 (SK1), was up-regulated specifically in the Aβ40 treated astrocytes, while displaying down-regulation in the Aβ42 astrocytes. SK1 promotes cytoprotective and substrate degradative autophagy of the huntingtin protein in Huntington’s disease [69], and its up-regulation here could indicate activation of substrate degradative autophagy induced by Aβ40 fibrils.

Similar to the astrocytes, comparative differential gene expression analysis revealed that a much greater number of up-regulated genes were observed in primary microglia treated with Aβ42 fibrils than those treated with Aβ40 fibrils, with 2133 and 251 occurring in each, respectively (Figure 5A). Among these genes, 249 were shared between the Aβ42 and Aβ40 fibril-treated microglia (Figure 5A). Thus, in microglia, Aβ42 fibril treatment resulted in nearly 10X the number of up-regulated genes compared with Aβ40 fibril treatment, and nearly all the Aβ40 fibril-induced changes also occurred in the Aβ42 fibril treatment. In order to highlight some of the important specific and common expression changes induced by the fibril treatments, we ranked the altered genes according to average abundance among all treatment groups, and relative expression of the 20 most abundant genes which were specifically elevated (≥5-fold control expression) in each Aβ fibril treatment, as well as those common to both. These are depicted in the heat maps in Figure 5B, along with the corresponding expression in the other treatment group. Notable genes among those specifically elevated in the Aβ42 fibril treatment are *Arg1*, *Cebpb*, and *f13a1*. *F13a1* up-regulation has been reported to be an indicating factor of microglia, adopting a macrophage-like phenotype, as seen in glioblastoma [70]. It is possible that Aβ42 fibrils can induce similar changes in microglia, as evidenced by f13a1 elevation. On the other hand, up-regulation of *Arg1* and *Cebpb* suggests a neuroprotective activation of microglia, as *Arg1* is a key marker of the proinflammatory M2a phenotype, and *Cebpb* is a reported transcriptional regulator of M2a genes, including *Arg1* [71,72]. Additionally, many of the commonly up-regulated microglial genes, including *IL1b*, *ccl2*, *cxcl1*, *Tnf*, *MMP9*, and *Nos2,* are proinflammatory markers [73,74,75,76]. Though our analysis indicated that two genes, *Fam110c* and *Cftr*, were specifically elevated by Aβ40 fibril treatment in the microglia. The data are suggestive that *Fam110c* and *Cftr* are also up-regulated by Aβ42 fibrils, though they failed to reach statistical significance in our study (*p* > 0.054 and 0.079, respectively). Therefore, Aβ40 fibrils did not induce significant unique enhanced gene expression changes in microglia compared to Aβ42 fibrils. 

### 3.4. Validation of Aβ Fibril Treated Glial Specific Genes in AD Rat brain

Next, we sought to validate the increased expression of reactive glial genes in the transgenic AD model, TgF344-AD rats [55]. We confirmed the presence of parenchymal amyloid plaques in 15-month-old F344Tg-AD rats which were primarily composed of fibrillar Aβ42 (Appendix A). The RNA sequence data indicated that *Gbp2* and *C3* genes (Figure 4B) and *iNOS* were significantly up-regulated in Aβ42 fibril-treated astrocytes. In accordance with our RNA sequence data, immunolabeling in the TgF344-AD rats brain revealed enhancement of iNOS, GBP2, and C3, particularly around amyloid plaque deposits in F344Tg-AD compared with WT rat brain, as well as strong co-localization with the astrocyte marker GFAP (Figure 6B,D,F, respectively). Thus, we confirmed increased expression of iNOS, GBP2, and C3 in astrocytes associated with amyloid plaques in AD rat brains. 

In the treated microglia, *Mmp9*, *Mmp12,* and *Ccl2* were significantly elevated in both Aβ42 and Aβ40 fibril-treated cells (Figure 5B), while *PAI-1/Serpine1* and *C1r* were specifically elevated in the Aβ42-treated microglia (counts of *PAI-1/Serpine1*: 366 vs. 4392; *C1r* 447 vs. 8706 in control and treated cells, respectively). PAI-1 is a reported regulator of motility and phagocytosis in microglia [77], while C1r, an important member of the classic complement cascade, is reported to be up-regulated in AD models upon plaque formation [78]. Based on the proinflammatory functions and relevance to microglia function in ADRD pathology, we sought to validate the enhancement of these markers in TgF344-AD rat brains. We found increased number of activated microglia expressing PAI-1, MMP9, MMP12, CCL2, and C1R surrounding plaques in F344Tg-AD rat brains compared with WT rat brains (Figure 7B,D,F,H,J, respectively). Collectively, the RNA sequence and immunolabeling data reveal a substantial increase in the expression of reactive glial genes in primary cultures treated with Aβ42 fibrils, as well as surrounding Aβ42 fibril plaque deposits, in the F344Tg-AD rat brain.

### 3.5. Ingenuity Pathway Analysis of Glial Cell Activation Mechanisms

In order to add mechanistic context to the observed gene expression changes, we utilized Ingenuity pathway analysis (IPA) to identify altered pathways, regulators, and downstream functions. Based on expression changes in downstream target molecules or interacting genes, IPA predicts activation (z score > 2) or inhibition (z score < −2) of upstream regulators, pathways, and disease functions [79]. Using IPA, we comparatively analyzed the up-regulated genes in the astrocytes treated with Aβ42 and Aβ40 fibrils. Interestingly, the disease function inflammatory response was predicted as activated in the Aβ42 fibril treated astrocytes (z = 2.744), with suggested activation (z = 1.585) in the Aβ40 fibril treated astrocytes, as shown in the heat map in Figure 8A. IPA analysis predicted the activation of the proinflammatory cytokine TNFα in the Aβ42 fibril (z = 3.892), but not the Aβ40 fibril treated cells. A heat map depicting the relative expression of the TNFα associated genes in both Aβ fibril treatments is shown in Figure 8B. Consistent with our other results, Aβ42 fibrils are predicted to have a greater proinflammatory effect on IPA, both on its activation of inflammatory cytokines, such as TNFα, and on its activation of the inflammatory response in astrocytes. 

Similarly, we comparatively analyzed the up-regulated genes in microglia treated with Aβ42 and Aβ40 fibrils. Due to the large number of up-regulated genes in the treated microglia, the IPA analysis was restricted to only genes with expression ≥10-fold of the control expression. IPA analysis indicated predicted activation of the downstream function activation of microglia in both Aβ42 fibril- (z = 3.204) and Aβ40 fibril- (z = 2.054) treated cells. Consistent with our other observations, the effect of Aβ42 fibril treatment was shown to be more robust for the activation of microglia in the pathway analysis. A heat map depicting the relative expression of genes associated with microglia activation in each Aβ fibril treatment group is depicted in Figure 8C. It is noteworthy that genes associated with both M1 and M2 polarization of microglia are present, primarily in the Aβ42 fibril treated cells, showing further evidence that Aβ fibril treatment induces both M1 and M2 activation of microglia, with Aβ42 fibrils displaying a greater ability to do so. IPA analysis also predicted activation of TGF-β1 in both Aβ42 fibril- (z = 4.404) and Aβ40 fibril-treated microglia (z = 2.118). A heat map depicting the relative expression of the most significantly up-regulated genes associated with TGF-β1 in each Aβ fibril treatment group is depicted in Figure 8D. It has been reported that TGF-β1 can mediate a reparative or alternative activation phenotype in microglia [80,81]. It is, therefore, possible that induction of TGF-β1 could contribute to the alternative microglia phenotype observed here in the Aβ fibril-treated cells. 

Because IPA indicated microglial activation in both the Aβ42 and Aβ40 fibril treated microglia, we decided to probe our data set for genes found in the Disease Associated Microglia (DAM) phenotype, including those both up-regulated or down-regulated [82,83,84,85,86]. Despite showing a more robust activation of microglia in the IPA analysis, and an overall greater impact on up-regulated genes, Aβ42 fibril treatment caused significant up-regulation of many DAM up-regulated genes, as well as down-regulation of 2 DAM down-regulated genes (Figure 9). This suggests that Aβ42 fibrils are triggering DAM phenotype with consistent microglial activation. Surprisingly, Aβ40 fibril treatment resulted in the down-regulation of most of these DAM-associated up-regulated genes, especially the key DAM-triggering receptor gene *Trem2*, while also causing the down-regulation of four DAM-associated down-regulated genes (Figure 9). This indicates that Aβ40 fibril treatment induces directional differential expression of microglial genes that both aligns and distinguishes them from DAMs. 

## 4. Discussion

Increasing evidence suggests that neuroinflammation is a key player in the pathophysiology of ADRD [15,87,88]. This process is marked by the reactive gliosis surrounding amyloid deposits in the brain, involving multiple distinct species of Aβ fibrils. For example, it involves the Aβ42 fibrils primarily comprising the hallmark parenchymal plaques in AD and the Aβ40 fibrils primarily associated with the vascular amyloid deposits observed in CAA [2,3,4,11,12,89]. As observed in several animal models and AD patients, chronic glial activation and pro-inflammatory cytokine and chemokine production stimulate neurodegenerative processes [87,90,91,92]. Importantly, overexpressed pro-inflammatory cytokines are not only present in early stages of clinical AD, but also contribute to neurotoxicity and cognitive deterioration, promoting severe late-stage AD [93,94,95]. Microglia and astrocytes are the primary neuroimmune cells involved in this process, but capillary endothelial cells and infiltrating blood cells also contribute to neuroinflammation [15,26,96]. Similarly, dyshoric amyloid projections in CAA drive dramatic perivascular inflammation associated with activation of both astrocytes and microglia as well [20,24,25]. The mechanisms responsible for the inflammatory pathogenesis of ADRD, including the activation of both astrocytes and microglia, as well as the specific contributions of distinct Aβ fibril species, are complicated and remain unclear. Therefore, the present study focused on comparatively investigating molecular actions in both astrocytes and microglial cells treated with well-characterized Aβ42 and Aβ40 fibrils that were free from non-fibrillar species. 

Previous studies which investigated the molecular impacts of Aβ fibril treatment on glial cells have largely incorporated single Aβ fibril species, or a combination of fibrils, with varying lengths, and have not comparatively investigated the specific effects of distinct fibril species [18,19,36,37,40,50,51,52]. Additionally, these studies have mostly explored either astrocytes or microglia, and rarely include comparative analysis of both cell types [18,19,36,37,40,50,51,52]. Thus, an important and novel feature of our study is the comparative analysis of the specific effects of distinct, homogeneous populations of both Aβ42 and Aβ40 fibrils on both astrocytes and microglia. 

RNA sequence analysis in astrocytes and microglia treated with Aβ42 or Aβ40 fibrils revealed that, in general, Aβ42 fibrils induced more gene expression changes in both astrocytes and microglia than Aβ40 fibrils, accounting for 30% more up-regulated genes in astrocytes and nearly 10 times as many up-regulated genes in microglia (Figure 4 and Figure 5). A notable, specific effect of Aβ40 fibril treatment in astrocytes was the unique enhancement of *Sphk1* (Figure 4B). *Sphk1* codes for sphingosine kinase 1, which is reported to promote protective autophagy of aggregated huntingtin proteins in Huntington’s disease [69]. It is possible that up-regulation of *Sphk1* induced by Aβ40 could be a mechanism of enhancement of autophagy of amyloid deposits in ADRD. Many of the common genes up-regulated in the astrocytes by both Aβ42 and Aβ40 fibrils, such as *C3*, *Cfb*, and *Serping1*, are associated with the proinflammatory and neurotoxic A1 specific phenotype, with the A1 specific *Gbp2* also up-regulated in the Aβ42 fibril treated cells (Figure 6A) [97,98,99]. Additionally, the inducible neurotoxic factor *Lcn2,* and the inducible nitric oxide synthase *iNOS*, which are both active in inflammatory responses of A1 astrocytes, were also commonly up-regulated in the treated astrocytes (Figure 4B) [68,100]. Immunolabeling confirmed the enhancement of iNOS, GBP2, and C3 proteins in astrocytes surrounding parenchymal amyloid plaques in the TgF344-AD model, which are primarily composed of Aβ42 (Figure 6). IPA analysis indicated activation of the proinflammatory cytokines TNFα and IL1β in Aβ42 fibril-treated astrocytes (z = 3.892, and 3.477, respectively), but not in Aβ40 fibrils, despite some commonly up-regulated genes being shared in those networks (Figure 8A,B). Both TNFα and IL1β have been suggested as important contributors, and, subsequently, potential therapeutic targets, for the severe neuroinflammation in AD pathology [101,102]. Therefore, they may have a central role in astrocyte mediated inflammation in ADRD. Together, these results indicate that both Aβ42 and Aβ40 fibrils induce proinflammatory activation of astrocytes, which is indicative of an A1 phenotype. However, the inductive effects of Aβ42 fibrils, both in the number of genes and the number of pathways impacted, were more significant. A1 astrocytes lose many normal astrocyte functions, including regulating neurotransmission, long-term potentiation, neuroenergetics, producing complement components, and decreasing excitatory neuronal function [97,103]. While A1 astrocytes have been characterized in the brains of AD patients, both A1-like and A2-like genes can be present, suggesting a diverse population of astrocytes in AD [104]. Nevertheless, our results here, namely the increased expression of potentially detrimental A1 genes, are supported by immunolabeling of specific proinflammatory markers in F344Tg-AD rats. Comparative pathway analysis suggests that Aβ fibrils are promoters of chronic astrocytic neuroinflammation in the brain, with Aβ42 fibrils harboring more robust pro-inflammatory properties than Aβ42 among astrocytes. 

Aβ42 and Aβ40 fibril-treated microglia displayed common enhancement of *IL1b*, *ccl2*, *cxcl1*, *Tnf*, *MMP9*, and *Nos2* (Figure 5B), all of which are pro-inflammatory markers [73,74,75,76] indicative of induction of the M1 microglia phenotype. Immunolabeling in the TgF344-AD rat brain, a model presenting with largely Ab42 fibrillar deposits, confirmed the enhancement of MMP9 and CCL2, along with the pro-inflammatory PAI-1, MMP12, and C1R in microglia surrounding parenchymal amyloid plaques (Figure 7). While these results indicate that both Aβ42 and Aβ40 fibrils induce the M1 phenotype in microglia, as was the case in the astrocytes, both in terms of affected genes and associated pathways, the effects of Aβ42 fibrils are much more robust. On the other hand, Aβ42 uniquely enhanced the expression of *Arg1*, *Cebpb*, and *f13a1*. *Arg1* and *Cebpb* are both markers of the M2a repair microglia phenotype, with CEBPB regulating the expression of many M2a related genes, including *Arg1* [71,72]. *Arg1* is also reported to contribute to wound healing and matrix deposition [105,106]. On the other hand, F13a1 is a marker of the M2 repair phenotype in macrophages, and has been reported as an indicator of microglia adopting a macrophage-like phenotype, which was observed in glioblastoma [70,107]. Thus, it is possible that ARG1, CEBPB, and F13a1 could contribute to clearance of Aβ in ADRD as part of an M2 phenotype induced by Aβ42.

IPA analysis indicated microglial activation in both Aβ42 and Aβ40 fibril-treated cells, prompting us to further investigate the relative expression of DAM-associated genes in these microglia. We compiled a list of DAM up-regulated and down-regulated genes from the reported literature [82,83,84,85,86] that were detected in our RNA sequencing analysis. Interestingly, Aβ42 fibrils caused up-regulation of numerous DAM up-regulated genes in our analysis, and down-regulation of two DAM down-regulated genes (Figure 9). This implies that Aβ42 fibrils promote a DAM phenotype in treated primary microglia, with its caused activation comprised mainly of the M1 and M2 associated genes discussed above. In contrast, Aβ40 fibril treatment resulted in a surprisingly significant down-regulation of 8/10 of these DAM up-regulated genes, as well as down-regulation of 4/4 DAM down-regulated genes. Thus, the directional differential expression induced by Aβ40 fibril treatment both aligns and differs from the reported DAM phenotype. It is possible that Aβ40 fibril treatment induces a unique subtype of the DAM phenotype specific to microglia, responding to Aβ40 fibril accumulation. Indeed, other subtypes of DAMs have been reported, including a transition from DAM1 to DAM2, which is possibly regulated by TREM2 [108], or white matter-associated microglia (WAMs), which is an activated microglia phenotype associated with aging that is prematurely induced in a mouse model of AD [109]. In summary, while both fibrils induce activation of primary microglia, Aβ42 and Aβ40 fibrils clearly induce unique expression profiles and DAM phenotypes.

## 5. Limitations of the Study and Future Directions

While both fibrillar Aβ40 and Aβ42 deposits accumulate in the ADRD brain, and both contribute to inflammatory- and glial cell-mediated responses, their potential individual distinct contributions to ADRD pathologies have yet to be determined. A limitation of the present studies described here is that they were restricted to responses of single glial cell types in culture, and, thus, do not replicate the diverse cellular population of the brain. In addition, the immunolabeling confirmation of selected markers was performed in an AD rat model of parenchymal plaques largely composed of fibrillar Aβ42. However, these novel findings warrant future investigations. For example, the impacts of the differentially expressed glial genes identified here are likely not confined to autocrine functions. Therefore, future studies should explore effects in co-culture systems, including neurons, in order to further elucidate the functional/pathological contributions of these changes. Evaluation of glial expression surrounding fibrillar Aβ40 deposits can also be informative. Finally, comparative administration of specific fibrillar Aβ species, either alone or in combination with rodent brains, could provide further insight into Aβ/disease associated phenotypes observed here, in which primary astrocytes and microglia are conserved in vivo in the presence of diverse cellular populations. In conclusion, this study presents significant novel findings regarding the stimulatory effects of two prominent fibrillar Aβ species associated with ADRD.

## Figures and Tables

**Figure 1 biomedicines-10-02982-f001:**
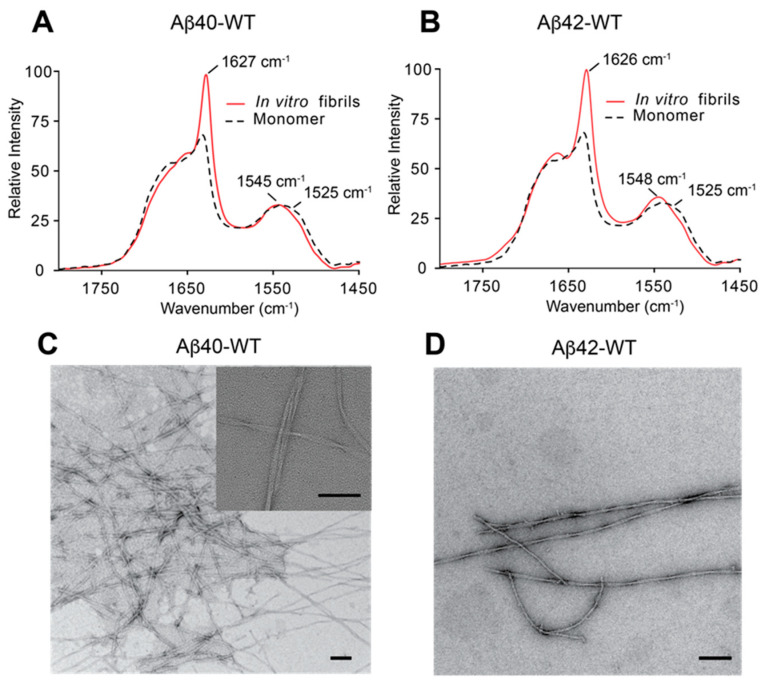
**Templated growth of Aβ40 and Aβ42 fibrils.** (**A**,**B**) The amide I and II regions in FTIR spectra of Aβ40 and Aβ42 are sensitive to the transition of Aβ monomers to fibrils. The Aβ monomers were incubated either with (red solid line) or without (black dashed line) fibril seeds. The amide I frequency of <1528 cm^−1^ and amide II frequency of >1545 cm^−1^ are characteristic of mature fibrils. (**C**,**D**) TEM micrographs of Aβ40 and Aβ42 fibrils after several rounds of template growth. The inset in (C) at higher magnification shows the twisted pairs of protofilaments in mature fibrils. Scale bars = 100 nm.

**Figure 2 biomedicines-10-02982-f002:**
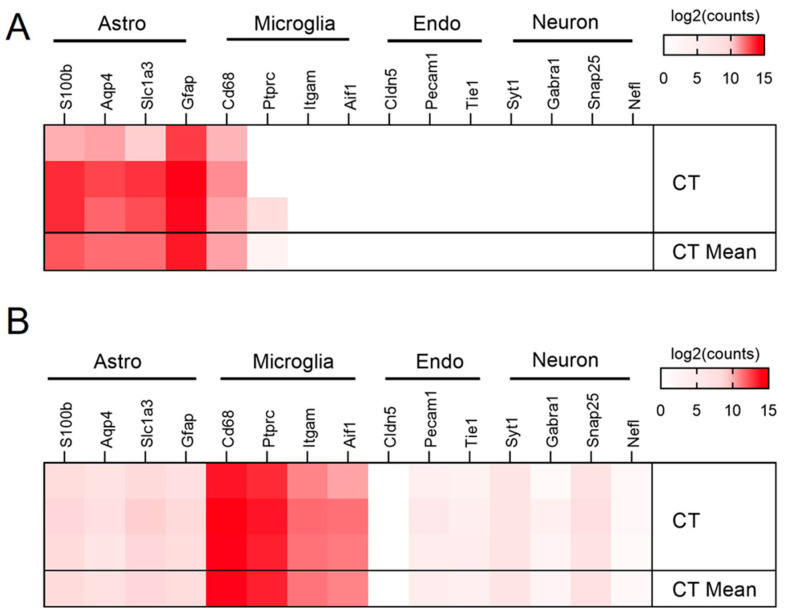
**RNA sequence validation of the primary astrocyte and microglia cultures.** Cell purity heatmaps depict gene counts for cell-type-specific markers for astrocytes (Astro), microglia, endothelial cells (Endo), and neurons in primary cultures of astrocytes (**A**) and microglia (**B**).

**Figure 3 biomedicines-10-02982-f003:**
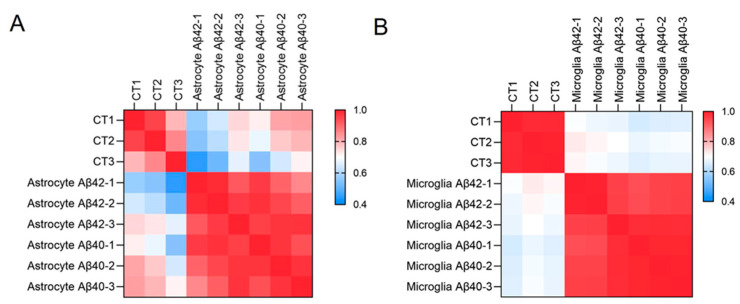
**Heat maps of a Pearson correlation between control, untreated cultures, and fibril treated cultures.** (**A**) astrocytes and (**B**) microglia. Data are calculated based on the whole gene counts from astrocyte and microglial RNA sequencing profiles, separately.

**Figure 4 biomedicines-10-02982-f004:**
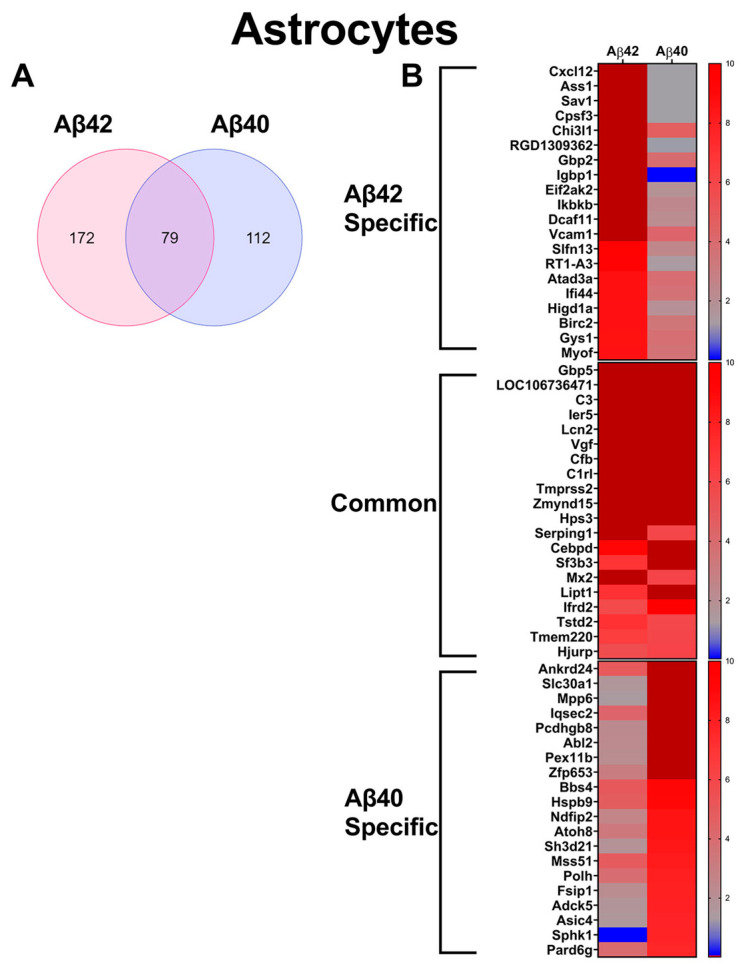
**Significantly increased genes in primary astrocytes treated with Aβ42 or Aβ40 fibrils.** (**A**) Venn diagram comparing significantly (*p* < 0.05) up-regulated genes, >5-fold of untreated controls, in primary astrocytes treated with Aβ42 or Aβ40 fibrils (*n* = 3). (**B**) Heat maps depicting relative expression of most enhanced genes specific to Aβ42 fibril treatment, common to both Aβ42 or Aβ40 fibril treatments and specific to Aβ40 fibril treatment, together with the corresponding expression in the other treatment groups.

**Figure 5 biomedicines-10-02982-f005:**
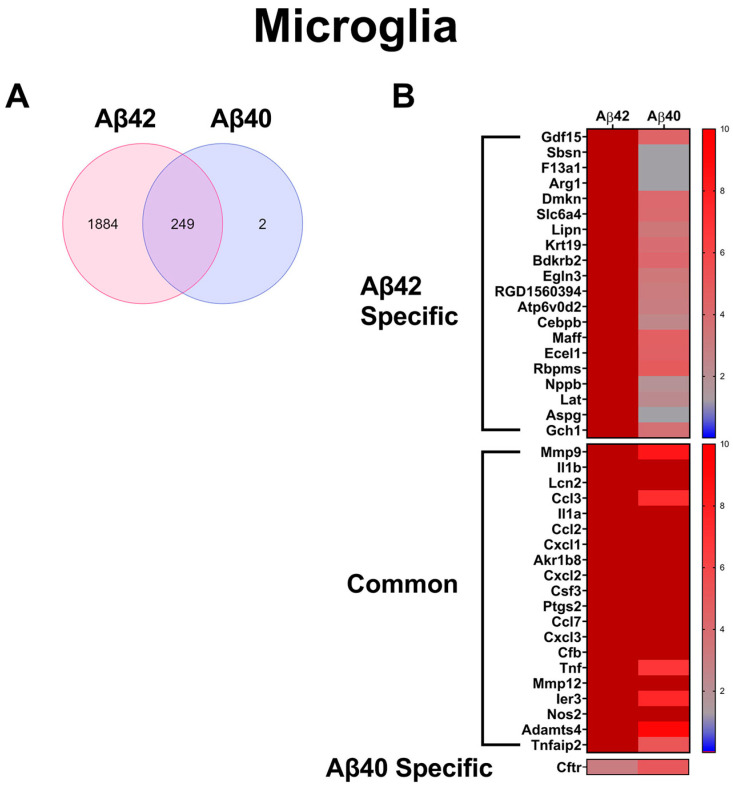
**Significantly increased genes in primary microglia treated with Aβ42 or Aβ40 fibrils.** (**A**) Venn diagram comparing significantly (*p* < 0.05) up-regulated genes, >5-fold of untreated controls, in primary microglia treated with Aβ42 or Aβ40 fibrils (*n* = 3). (**B**) Heat maps depicting relative expression of most enhanced genes specific to Aβ42 fibril treatment, common to both Aβ42 or Aβ40 fibril treatments and specific to Aβ40 fibril treatment, together with the corresponding expression in the other treatment groups.

**Figure 6 biomedicines-10-02982-f006:**
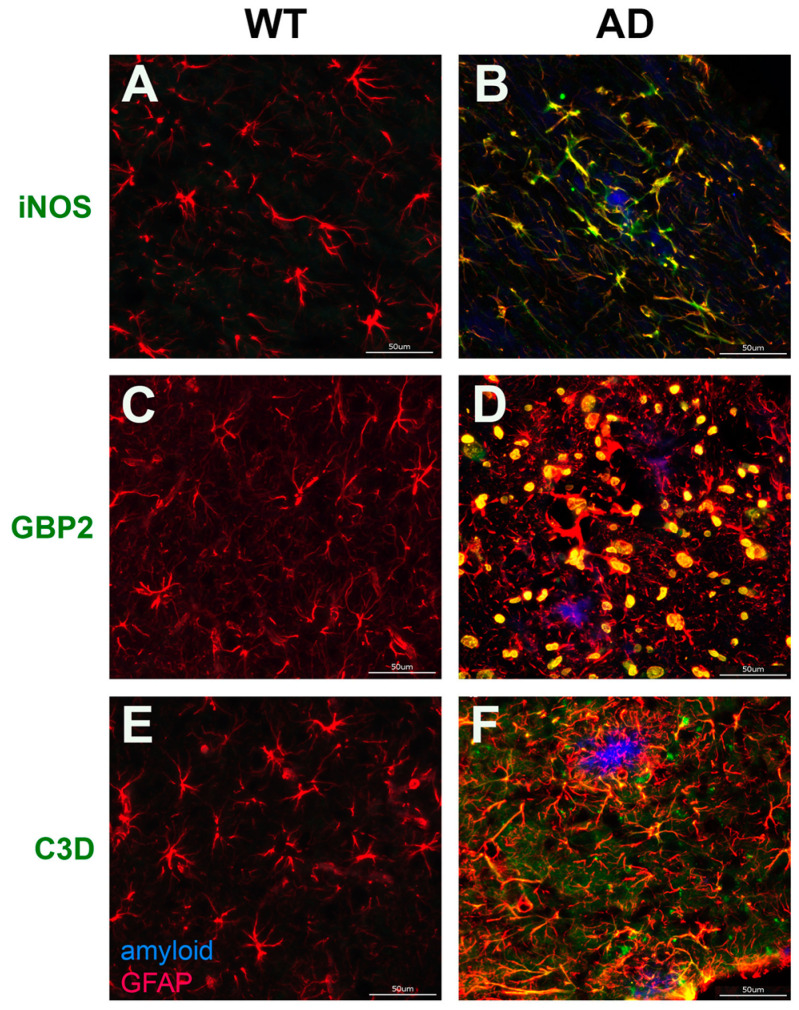
**Validation of Aβ42 fibril induced astrocyte genes by immunolabeling.** Increased immunolabeling for iNOS, GBP2, and C3D co-localized with astrocytes surrounding amyloid plaques. Brain sections from 15 M WT rats (**A**,**C**,**E**) and F344Tg-AD rats (**B**,**D**,**F**) were labeled with mAb66.1 to detect amyloid plaques (blue) and goat polyclonal antibody to GFAP for astrocytes (red), as well as rabbit polyclonal antibodies to iNOS, GBP2, and C3D (green). Scale bars = 50 µm.

**Figure 7 biomedicines-10-02982-f007:**
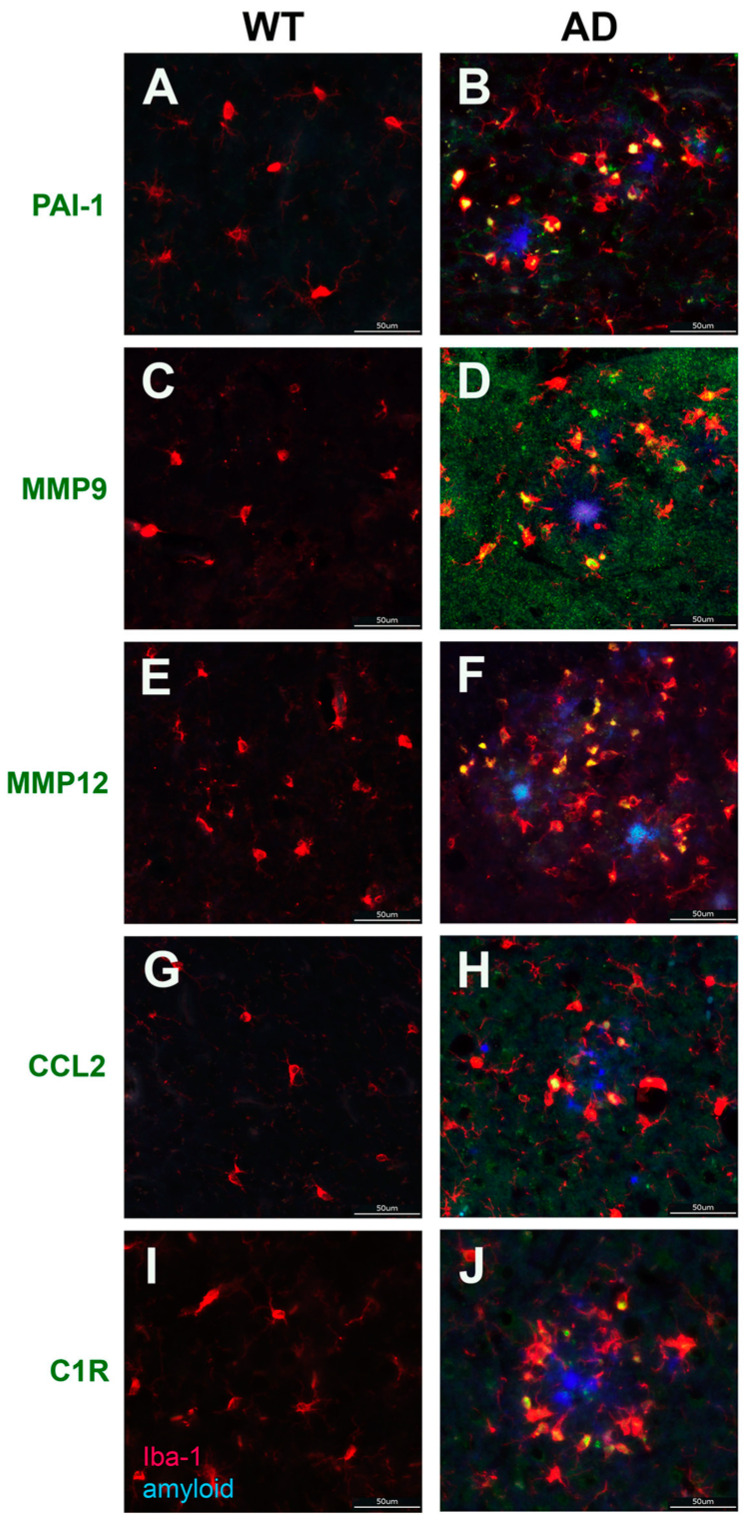
**Validation of Aβ42 fibril-induced microglia genes by immunolabeling.** Increased immunolabeling for PAI-1, MMP9, MMP12, CCL2, and C1R co-localized with microglia surrounding amyloid plaques. Brain sections from 15 M WT rats (**A**,**C**,**E**,**G**,**I**) and F344Tg-AD rats (**B**,**D**,**F**,**H**,**J**) were labeled with mAb66.1 to detect amyloid plaques (blue) and goat polyclonal antibody to Iba1 for microglia (red), as well as rabbit polyclonal antibodies to PAI-1, MMP9, MMP12, CCL2, and C1R (green). Scale bars = 50 µm.

**Figure 8 biomedicines-10-02982-f008:**
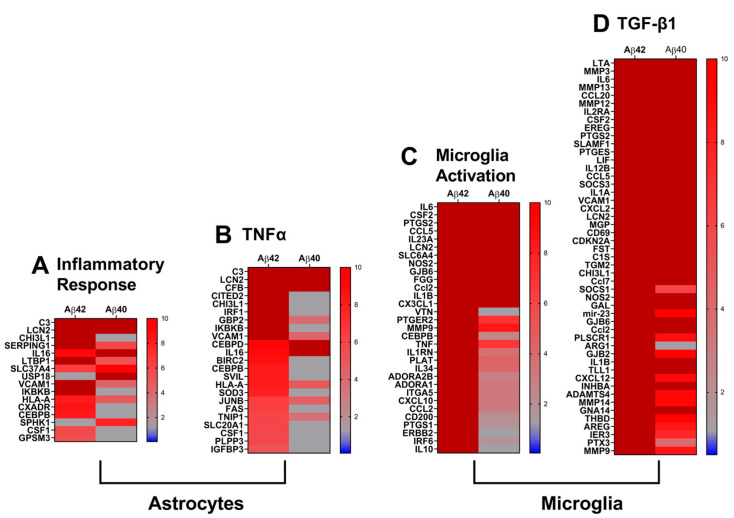
**IPA analysis of elevated signal pathways and involved genes.** Heat maps depict relative expression of genes in astrocytes associated with (**A**) inflammatory response; (**B**) TNFα, and of genes in microglia associated with (**C**) microglial activation and (**D**) TGF-β1. Red indicates significantly increased expression; blue, significantly decreased expression; grey, no significant change. Color intensity is relative to the degree of change.

**Figure 9 biomedicines-10-02982-f009:**
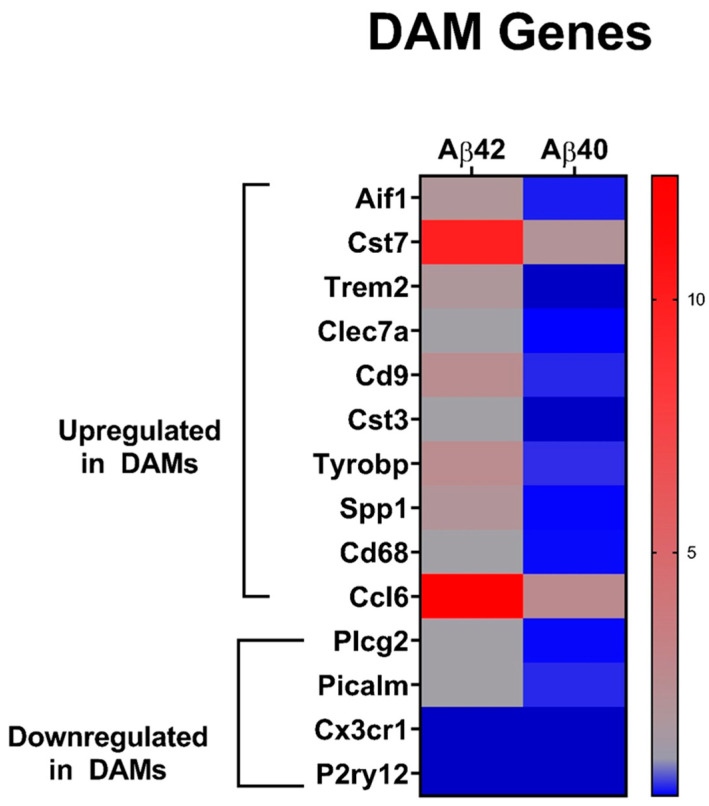
**Relative expression of DAM genes in Aβ42 and Aβ40 fibril-treated primary microglia.** Heat map depicting the relative expression (expression ratio compared to control microglia) of DAM-associated genes in Aβ42 and Aβ40 fibril-treated microglia. Red indicates significantly increased expression blue, significantly decreased expression; grey, no significant change. Color intensity is relative to the degree of change.

## Data Availability

Raw RNA sequencing data can be found in the NCBI GEO Datasets (https://www.ncbi.nlm.nih.gov/gds/), GEO submission ID# GSE211198.

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
