# Peer review of "Impact of Aβ40 and Aβ42 Fibrils on the Transcriptome of Primary Astrocytes and Microglia"

_biomedicines, 2022, doi:10.3390/biomedicines10112982_

Round 1

Reviewer 1 Report

The Authors presented a solid in vitro/in vivo experiment to understand Alzheimer's disease pathophysiology.

The proposed goal is a novel approach to studying single cell (astrocyte and microglia) contribution under controlled fragment-induced amyloid beta fibrils.

Though the idea is well developed, in my personal opinion, does not add significant knowledge to Alzheimer's disease pathophysiology. In fact the single challenge of isolated mono-fragment fibrils is substantially not mimicking any aspect of the human pathology, where these controlled environments are impossible. Moreover, adding preformed fibrils is not a novel approach and other cited manuscripts avoided the segregation of 40-42 Abeta fragments also to reduce this kind of pitfall. 

All that considered the in vivo model does not replicate totally the in vitro experiment. Other animal models are more apt to express the two fragments of the protein.

Authors should enlist these considerations as limitations of the experimental setting.

Minor:
The statement "Astrocytes are involved in regulation of cerebral blood flow, maintenance of synaptic homeostasis, and neurotrophic support for synapses" is partly redundant and not supported by citation 25. I suggest: (Zhu, W. M., et al. (2022). Neurovascular coupling mechanisms in health and neurovascular uncoupling in Alzheimer's disease. Brain : a journal of neurology145(7), 2276–2292. https://doi.org/10.1093/brain/awac174; Virtuoso, A.et al., The Spatiotemporal Coupling: Regional Energy Failure and Aberrant Proteins in Neurodegenerative Diseases. International journal of molecular sciences22(21), 11304. https://doi.org/10.3390/ijms222111304; and De Luca, C., et al., Roadmap for Stroke: Challenging the Role of the Neuronal Extracellular Matrix. International journal of molecular sciences21(20), 7554. https://doi.org/10.3390/ijms21207554). The same can be said, minorly, about references 26 and 27.

Line 69. What does "mediating neuronal network structures" mean?

Line 72 and 81 eliminate ]; line 91 eliminate ..; line 133 should be Aβ42; line 135 is redundant

For readability, the following paragraphs should be rephrased "Under pathological challenge, both astrocytes and microglia are recognized to be activated to contribute to neuroinflammatory processes"; "Confirming the in vivo relevance of these results, immunolabeling confirmed the enhancement of iNOS ..."

Reviewer 2 Report

The manuscript, biomedicines-1972129: Impact of Abeta40 and Abeta42 fibrils on the transcriptome of primary astrocytes and microglia, by Zhu X et al, focuses on understanding the effect of Abeta40 or Abeta42 fibrils on the astrocytes and glial cells from a gene expression standpoint. The authors generate homogenous (single strain) fibrils from Abeta40 or 42 and treat the neuronal cells to mimic Abeta plaque deposition in the brains of Alzheimer’s disease (AD) patients. Bulk RNA sequencing is utilized to study differences in gene expression patterns upon exposure to Abeta40 or 42 fibrils versus no fibrils (control). Several genes related to various cellular pathways including autophagy, chemokines production, and inflammation is seen differentially expressed in different cell types and upon Abeta40 or Abeta42 fibrils treatment. AD rat model tissue immunolabeling is used to validate the increased expression of glial or astrocyte genes upon fibril treatment. Authors conclude from this study that Abeta42 fibrils have more stimulating action on glial cells compared to Abeta40 fibrils treatment. While the Amyloid versus Oligomer hypothesis still remains a debate in the amyloid field, the results from this work mostly support the former and are extremely important from a clinical perspective. As the authors rightly point out, designing therapeutics will require deciphering the underlying mechanisms of disease pathogenesis. The manuscript is well-written and the experiments are designed and executed in a systematic manner. While overall appreciating the work, the reviewer also notes there are certain areas in the manuscript that needs improvement and other parts that will require additional studies to substantiate or strengthen the current findings. Please find the relevant questions and comments below.

1) This may sound philosophical, but it is a highly recommended experiment also: Having unraveled several upregulated and downregulated genes, the reviewer suggests the authors perform at least one functional assessment/assay to precisely reveal the cytotoxic mechanisms of Abeta fibrils. The authors, for instance, state that Sphk1 upregulation induced by Abeta could trigger autophagy. Using cultured neuronal cells or the ones already used in the present study, it is possible to design experiments to test if this is true under the conditions being tested (i.e. exposure to fibrils). Likewise, TGFbeta1 /Wnt signaling is another potential pathway to look at. Given this pathway is implicated in various cellular processes, how exactly is TGFbeta1 activity affected and what are the downstream consequences upon Abeta fibril accumulation? Finally, the authors could choose one or two top genes that are influenced by Abeta40 as well as Abeta42 (classified as “common”) and ask the question of how closely these gene-encoded protein functions are altered in the cells exposed to the fibrils by performing suitable biochemical studies. The reviewer insists on these ideas because it is the drug development either against the Abeta fibrils/oligomeric structures or the genetic modifiers/cellular pathways that will dictate the future of AD therapy. Exploring a minimum of one example at a time from a list of hundreds of genes discovered from this extensive study would be a start. Including this would make the manuscript stronger besides motivating RNA-seq users and experts to explore beyond what they learn from transcriptome analysis.

2) The method of Abeta40 fibril preparation is different compared to that of Abeta42. Though it is believed that Abeta42 fibrils exert relatively higher cytotoxicity compared to Abeta40 fibrils, the differential effect of the fibrils on gene expression observed in the same cell types could be a result of the difference in fibril preparation protocol. 1) From the point of view of the current study, what would be the consequence of preparing quiescent Abeta42 fibrils or generating Abeta40 fibrils by agitation, and then exposing these to cells? 2) Which of these (quiescent versus agitated Abeta40 or Abeta42) are physiologically appropriate and why? Please explain or support with experimental evidence.

3) Please label the peaks on the 2D DARR NMR spectrum in Supplemental Figure1, A, and C  

4) There were figures mislabeled in the text in more than one instance. Please go through the manuscript's main text carefully and fix these problems; otherwise, it becomes difficult for the reader to follow.

5) The authors write in the methods that for both Abeta40 and Abeta42 templated-fibrillation, the monomers used were Abeta40, which probably is an error. Please correct this.

6) Please ensure the correctness of all the references cited in the manuscript.

Round 2

Reviewer 2 Report

In the revised manuscript, the authors have addressed most of the reviewer's questions. The manuscript may be accepted in its current form.